# Analysis of the Effect of Fe_2_O_3_ Addition in the Combustion of a Wood-Based Fuel

**DOI:** 10.3390/ma15217740

**Published:** 2022-11-03

**Authors:** Jerzy Chojnacki, Jan Kielar, Waldemar Kuczyński, Tomáš Najser, Leon Kukiełka, Jaroslav Frantík, Bogusława Berner, Václav Peer, Bernard Knutel, Błażej Gaze

**Affiliations:** 1Faculty of Mechanical Engineering, Koszalin University of Technology, Racławicka Str. 15-17, 75-620 Koszalin, Poland; 2Centre of Energy Utilization of Non-Traditional Energy Sources—ENET Centre, VSB—Technical University of Ostrava, 17. Listopadu 2172/15, 708 00 Ostrava, Czech Republic; 3Institute of Agricultural Engineering, Wrocław University of Environmental and Life Sciences, 51-630 Wrocław, Poland

**Keywords:** catalyst, iron oxide, boiler, biomass, combustion, emissions

## Abstract

A comparative study was carried out of emissions from the catalytic combustion of pellets made from furniture board waste and pellets made from wood mixed with Fe_2_O_3_. The mass content of the Fe_2_O_3_ catalyst in the fuel was varied from 0% to 5%, 10%, and 15% in relation to the total dry mass weight of the pellets. The average flame temperature in the boiler was between 730 and 800 °C. The effect of the catalyst concentration in the fuel was analysed with respect to the contents of O_2_, CO_2_, CO, H_2,_ and NO_x_ in the flue gas and the combustion quality of the pellets in the heating boiler. Changes in the CO_2_ content and the proportion of unburned combustible components in the combustion residue were assessed. It was established that an increase in the Fe_2_O_3_ content of the prepared fuels had a positive effect on reducing NO_x_, CO, and H_2_ emissions. However, the proportion of iron oxide in the tested fuel pellets did not significantly influence changes in their combustion quality. A strong effect of the addition of Fe_2_O_3_ on the reduction of the average NOx content in the flue gas occurred with the combustion of furniture board fuel, from 51.4 ppm at 0% Fe_2_O_3_ to 7.7 ppm for an additive content of 15%. Based on the analysis of the residue in the boiler ash pan, the amount of unburned combustibles relative to their input amounts was found to be 0.09–0.22% for wood pellets and 0.50–0.31% for furniture board waste pellets.

## 1. Introduction

Environmentally friendly biomass, replacing fossil energy carriers as a fuel, is becoming increasingly important for the health and safety of society and for economic growth. What is becoming particularly important is the energy management of waste containing organic substances from agricultural and forestry production, the processing industry, and human society [1]. Based on its water content, biomass is divided into wet and dry biomass [2]. Wet biomass is mainly organic materials with such a high water content that they are unsuitable for direct combustion. However, they may be converted into gaseous fuel in biogas plants [3]. Dry biomass wastes are most frequently pelleted or briquetted to stabilise their physical parameters. Biofuel briquettes and pellets differ in many respects from conventional solid fuels, such as coal or coke, due to higher moisture content, lower calorific values, and the different content of additional components such as chlorine, sulphur, phosphorus, nitrogen, and metals, that they contain. These factors can affect the content of the emission components of the gases produced during their combustion and the properties of the ash [4]. Thus, these particular characteristics of biomass fuels cause many challenges; at the same time, in many cases, they also provide benefits.

Commercially available solid fuel for combustion in domestic boilers is mainly sold in the form of pellets, which are produced chiefly from wood or waste containing wooden mass. Cereal straw and typical energy crops such as miscanthus are also used in large quantities for pellet production [5,6]. Any material amenable to pressure agglomeration, such as tobacco post-harvest waste, can also be used for this purpose [7]. To improve the mechanical parameters of pellets made from straw, natural binders are used, such as, for example, flour or waste biomass from food industry production [8,9,10].

Biomass combustion in heating boilers can contribute to the emission of excessive amounts of substances that are harmful to humans and the environment, such as particulate matter (PM), carbon monoxide (CO), nitrogen oxides (NO_x_), and sulphur oxides (SO_x_) [11,12]. Polycyclic aromatic hydrocarbons (PAH) and volatile organic compounds (VOC) are also emitted in the flue gas [13]. The fuel composition mainly influences emissions but can also be affected by the combustion method, flue gas recirculation, and the boiler air supply [14,15,16]. At the same time, the shape of the combustion chamber or the support of biomass combustion with gaseous fuel may reduce the number of emissions produced in the boiler [17].

Pellets made from waste biomass may also present problems during combustion due to the quality and composition of the resulting ash, e.g., thermal sticking of the ash may occur, causing clogging of the grate [7,12,15,18]. One of the more common methods to reduce the environmental costs of biofuel combustion is the use of catalysts. These can be placed in the boiler flue gas lines [19] or as catalytic admixtures to the solid biomass. Catalysts in this form are used to improve the combustion quality by accelerating the oxidation process and to improve the ash quality and flue gas [20,21,22,23,24]. In the combustion of fuels produced from biomass, attempts have been made to use metal compounds as catalysts, for example, iron, calcium, or aluminum oxides, as well as vanadium, magnesium, copper, and titanium oxides as sodium and lithium chlorides [20,25].

Research into the use of iron oxides as catalytic additives has been pursued due to their low cost, environmental friendliness, high reactivity, and susceptibility to accessible surface and structural modifications. Three forms of iron oxide are known to occur: (III) Fe_2_O_3_, which occurs in nature as iron ore and most commonly as haematite containing approximately 70% pure iron, iron (II) FeO oxide, which infrequently occurs as a mineral-its mineral form is known as wüstite, and iron oxide (I) Fe_3_O_4_, which occurs in nature as magnetite, so-called because of its strong magnetic properties [26]. Iron oxides are used to improve the combustion quality of many fuels and combustible substances, for example, in thermite. Iron oxide nanopowders are used as a catalyst in the burning of solid composite rocket fuels, as well as for the oxidation of soot formed during the combustion of liquid fuels and plastics [27,28,29,30]. Iron oxide in the form of Fe_2_O_3_ can accelerate the combustion process by being an oxygen carrier [31]. Materials consisting of iron oxides in combination with another metal can also be used as catalysts to assist in the oxidation of solid fuels or gases. Au/Fe_2_O_3_ catalysts prepared using the co-precipitation method positively affect CO oxidation at low temperatures, and assist the combustion of volatile organic compounds [32,33]. Attempts have also been made to use iron oxides together with minerals, for example, Fe_2_O_3_-coated olivine ((Mg,Fe)_2_[SiO_4_]), as a catalyst in the gasification of biomass alone and co-gasification of biomass with coal to crack/reform tar and to increase H_2_ yield [34]. The action of iron oxide and copper as a catalyst in the thermal processing of biomass is also known [35].

Carbon monoxide is used as a feedstock for iron smelting using coke. In the presence of burning coal, iron oxides are reduced with carbon, carbon monoxide, and carbon dioxide. The reduction starts as early as 300 °C and continues until all the carbon is consumed or escapes as gases [36]. Further heating in the presence of oxygen can re-oxidise the reduced iron by diffusing oxygen from the surrounding atmosphere and causing it to return to its previous form [37]. It has been found that iron oxide can also be reduced by biomass, and the reduction is temperature dependent and strengthened at temperatures above 1100 °C [38]. In the case of biomass, iron oxide reduction is also enabled by carbon and hydrogen, which contain materials of organic origin. Research to date has been conducted on a micro-scale in many cases and has not included an analysis of solid biomass combustion in boilers as a fuel mixed with Fe_2_O_3_.

The aim of the study was to verify the hypothesis of whether the use of Fe_2_O_3_ iron oxide as a catalytic additive for biomass fuel has an impact on reducing the resulting fuel combustion emissions of NO_x_, CO, and H_2_. An additional aim was to establish whether adding Fe_2_O_3_ to biomass could improve the quality of fuel combustion in the boiler by causing better after-combustion of combustible substances contained in the fuel. An assumption was accepted that, during combustion, there would be a reduction of oxygen from Fe_2_O_3_, presuming that pure iron would be produced from a part of the catalyst, and the free oxygen would assist the biomass combustion process by reducing the amount of oxygen taken from the air.

## 2. Materials and Methods

### 2.1. Materials

Pellets made from shredded wood-based furniture boards laminated with white film were used as the base biofuel. Urea-formaldehyde and melamine-urea resins used to glue the layers of furniture boards together, as well as polyurethane adhesives and melamine-urea-phenol-formaldehyde resins, constitute a potentially good source of NO_x_ and HCN emissions during combustion [39]. Furniture board waste was collected from a kitchen furniture manufacturing company. The material was ground using a hammer mill, and then the moisture of the material was determined in the resulting loose mass. The substrate was divided into four equal parts. The first part was left without any addition of the catalyst, while the catalyst was added to the other parts so that the catalyst content in each part was 5%, 10%, or 15% of the total dry weight of the resulting material. The entire material prepared was pelletised.

The catalyst, i.e., iron oxide Fe_2_O_3_, was purchased as a fine powder with a minimum purity of 99.00%. The average grain size of the catalyst ranged from 1.0 to 2.6 µm. Its density was 5.24 g·cm^−3^, its bulk density was 0.48 g·cm^−3^, and the determined moisture content was 0.25%.

For comparative purposes, a second type of pellets made from A1 grade spruce wood was prepared as a base biofuel. For this purpose, finished wood pellets made from this raw material were purchased; these were then chipped and the moisture content of the material obtained was determined. Finally, as in the case of chipped furniture boards, four types of pellets were produced, with catalyst content of 0%, 5%, 10%, or 15%, calculated according to the total dry weight of the material obtained. The choice of an extensive range of Fe_2_O_3_ proportions in the prepared fuels was based on the assumption that reduced oxygen from iron oxide would assist the combustion process and on the lack of available knowledge on the catalytic effect of iron oxide in reducing gas emissions from biomass combustion. The diameter of all pellets prepared was 6 mm. Examples of the produced granulate, without and with catalyst content, are shown in Figure 1.

### 2.2. Methods

#### 2.2.1. Fuel and Residues after Combustion Assessment Methods

An analysis of the elements contained in the fuel was performed solely on the basic versions of the fuel (0% Fe_2_O_3_ addition). A CHSN628 analyser manufactured by LECO was used for this purpose. The determination of other physicochemical properties, such as volatile and solid fuel content and moisture and ash content, was performed using the TGA (Thermogravimetric Analysis) method with a TGA 701 analyser. The proportion of the gaseous fuel was determined in the thermogravimeter until the temperature of 430 °C was reached. The analyses were repeated three times. The chemical content of the fuel was also determined using an AAS contrAA^®^ 700 analyzer from Analytik Jena GmbH (Jena, Germany). The moisture content of all the fuels prepared was also determined using the dryer method, separately for each biofuel type and each Fe_2_O_3_ range, according to PN-N ISO 18134-1:2015-11.

By analysing the proportion of unburned combustibles in the ashes from the combustion of the prepared materials, an attempt was made to assess the influence of the Fe_2_O_3_ content in the pellets on the combustion quality of the fuel. The residues from the combustion of each fuel, with and without all Fe_2_O_3_ contents, were used for the study. Moisture content and unburned volatile and solid fuel content were determined. In addition, the chemical compounds found in the incinerated residues were tested. An AAS contrAA^®^ 700 analyser was also used for this purpose. The summed values of the volatile fuel and solid fuel contained in the ash residue, and their ash content as non-combustible minerals, were used to determine the ratio of the unburned fuel portion to the fuel input used to create it. This was done by determining the necessary amount of fuel input, based on the analysis of their composition and Fe_2_O_3_ content, to obtain a composition after combustion identical to the results obtained from the analysis of the ash residue.

#### 2.2.2. Flue Gas Emission Test Bench

The pellets were burned in an energy-efficient retort boiler with a nominal output of 31.5 kW (VARIANT SL-33 A, Slokov, Moravský Písek, Czech Republic). The boiler was equipped with a controller to regulate the operation of the conveyor and the air supplied to the combustion chamber. A diagram of the emission measurement test bench with the boiler and the measurement instrumentation is shown in Figure 2.

The boiler (1) was fired up and heated with a typical wood-based fuel; then, the fuel tested was introduced into it. To be able to observe and accurately dose the selected fuel type, a transparent calibrated pipe (5) was installed inside the fuel tank in the boiler, through which the fuel tested was delivered to the screw conveyor (9), pushing it into the burner (2). The pipe made it possible to determine the time of the onset of the experimental fuel uptake by the screw conveyor.

Three thermocouples are located in the boiler’s combustion chamber to allow a complete analysis of the temperature inside the flame. There was one thermocouple (4) to determine the temperature value in the flue gas exhaust pipe from the boiler. Figure 2 shows the location of the thermocouples and the gas and temperature analyser probes. The temperature from each of the thermocouples used and installed in the boiler combustion chamber and the flue gas outlet pipe was measured and recorded using a Graphtec GL-840 datalogger. Based on the data from the three thermocouples in the combustion chamber, one average combustion temperature and its standard deviation were determined for further analysis.

Using a Wöhler A 550 analyser (Bad Wünnenberg, Germany), connected to a probe (3) placed in the flue gas outlet pipe from the boiler, the flue gas composition was measured. The measurement was carried out once the combustion process had been stabilized, every second throughout the combustion of the selected fuel type. The specifications of the Wöhler A 550 analyser can be found in Table 1.

The order in which the prepared fuels were selected for combustion in the boiler was done randomly so as not to burn the same material in succession with successive catalyst contents. The rotational speed of the fuel feeder screw was always constant, regardless of the fuel type. Its value was deliberately selected so that a certain amount of unburned fuel remained in the resulting ash from pellet combustion without the catalyst.

Changes in the combustion temperature in the boiler chamber and the flue gas temperature values in the chimney, as well as NO_x_, CO, H_2_ emissions, and O_2_ and CO_2_ contents in the flue gas, were recorded continuously. Once the tests had been completed, the values of these quantities were statistically processed using Statistica ver. 13.3 from StatSoft. Using the ANOVA analysis of variance, the significance of the effect of the iron oxide content in the tested fuel on the composition of the flue gases emitted and on the boiler combustion temperature values was determined.

Determining the effect of the Fe_2_O_3_ content in pellets on the combustion quality of the fuel was done primarily by comparing the results of the CO_2_ content in the flue gases from the combustion of all types of pellets, including pellets containing no Fe_2_O_3_ with the theoretically calculated carbon dioxide content in the flue gases of these fuels. For this purpose, the theoretical carbon dioxide content from the combustion of the fuel without a catalyst was calculated after taking into account its lower actual content in the pellet, which was due to the significant proportion of Fe_2_O_3_. In performing the calculations, the basic elemental composition of the materials produced for combustion was considered, including their catalyst content. It was assumed that the amount of carbon dioxide emissions would solely depend on the amount of fuel in the pellet and its chemical composition. A comparison of the actual CO_2_ content of the flue gas with the theoretically calculated one was used to check whether the combustion quality of the fuel changes with higher catalyst content.

To determine the catalysed effect of Fe_2_O_3_ on CO, H_2_, and NO_x_ emissions, it was assumed that their contents in the flue gas are derived from the amount of fuel actually burnt at that moment and that this value is related to the amount of CO_2_ actually emitted. The CO, H_2_, and NO_x_ content of the flue gas was therefore divided by the CO_2_ content emitted simultaneously. Comparison of the thus-determined emission values of these gases obtained without iron oxide with emissions with it as a fuel additive could indicate the catalytic properties of Fe_2_O_3_.

## 3. Results and Discussion

### 3.1. Fuel Analysis

Prior to the tests related to the combustion and determination of flue gas components, the pellet density and elemental composition of the biofuel were analysed. These tests were carried out on the base fuels only, without any addition of the catalyst: pellets made from wood–denoted as WP (Wood Pellets) in the following descriptions, and pellets made from furniture boards–denoted as FBP (Furniture Board Pellets). The density of the dried wood pellets was determined as 1.31 g·cm^−3^, and that of the furniture board pellets was 1.25 g·cm^−3^. The averaged results of the carbon, hydrogen, nitrogen, sulphur, and oxygen contents and the combustion heat value, converted to the dry mass of the prepared products, are shown in Table 2. The ash, volatile and solid fuel contents, and combustion heat values are shown in Table 3. The sum of volatile and solid fuels represented the organic mass contained in the biofuel.

The moisture content of all the prepared fuels was also determined using the dryer method, for each type of biofuel and each Fe_2_O_3_ content, in accordance with PN-PN ISO 18134-1: 2015-11. The results of the pellet moisture content are presented in Table 4. The higher moisture content of the pellets with the catalyst was due to the technological conditions of the pellet preparation with iron oxide.

### 3.2. Emission Analysis

The combustion of all the fuels in the boiler was carried out at the same rotational speed of the screw conveyor feeding the fuel to the retort burner. Using the data from the Wöhler A 550 flue gas analyser, graphs were prepared of the dependence of the contents of oxygen, carbon dioxide, hydrogen, carbon monoxide, and nitrogen compounds in the flue gas on the proportion of the catalyst in the fuel. The graphs are shown in Figure 3 and Figure 4.

To illustrate the results, the box plot of the values where the transverse line represents the mean value, the upper and lower border of the box are the ± value of the standard deviation, and the whiskers represent the minimum and maximum values obtained in the measurements. In order to facilitate a comparison of the results, the graphs of the same emissions from the combustion of wood pellets and furniture board pellets are shown side by side. The graphs show the trend lines determined from all the data collected from the measurement. For these lines, the linear equations of the dependence of the emission from the catalyst content of the fuel were determined.

The value of the p-coefficient was also calculated to determine the significance of the impact of the catalyst content on the emission content. The regression equations and the values of the p-coefficient are presented in Table 5.

By analysing the graphs in Figure 3 and Figure 4, it can be found that the level of emissions in the flue gases, CO_2_, H_2_, CO, and NO_x_, in the combustion of furniture board pellets compared to the emission level of these substances in the burning of wood pellets was always higher. The high nitrogen oxide emissions produced during the combustion of furniture board pellets were the result of the high nitrogen content of this raw material (Table 2). The results in Figure 3 indicate that there was always an excess of oxygen during the combustion of each granulate, resulting in low levels of CO emissions (Figure 4).

Any disruption in the combustion process manifested itself with increased CO and H_2_ emissions, as seen in Figure 4, at 5% Fe_2_O_3_ content during the combustion of furniture board pellets (FBP). The carbon dioxide content during the combustion of wood pellets and furniture board pellets was coordinated with the oxygen content in the flue gases. A decrease in the carbon dioxide content of the flue gas was associated with an increase in the oxygen content. A higher catalyst content in the fuel stimulated a reduction in the carbon dioxide content and a growth in the oxygen content. The critical level was the 10% Fe_2_O_3_ content in the fuel because, beyond this value, the amount of carbon dioxide in the flue gas started to increase, and the oxygen content decreased. Statistical analysis demonstrated a decrease in the CO content in the boiler flue gases and the significance of the effect of an increase of the catalyst content in the fuel on reducing CO content. As can be seen from the graphs, the decrease in carbon monoxide content in the gases was not very pronounced; very frequently, its content values did not increase despite an increase in the catalyst content.

The regression equations for the dependence of oxygen and carbon dioxide contents from the catalyst content in Table 5 indicate that only oxygen contained in the air was involved in the combustion process. The coefficients at “x” describing the effect of the Fe_2_O_3_ content in the fuel on the oxygen and carbon dioxide contents when combusting the same fuel are almost identical, which confirms that oxygen extracted from the chemical decomposition of the catalyst did not participate in the combustion process.

An analysis of the carbon dioxide content in the flue gas was carried out to verify whether the catalyst content in the pellets affects the quality of pellet combustion. A comparison was made between the CO_2_ contents obtained in the tests (Figure 3) and the values determined by stoichiometric equations from the elementary composition of raw fuel. The computations were based on the CO_2_ content in the flue gas from pellets with no catalyst. In the comparative calculations, the identical amounts of biomass were accepted as those in pellets with the addition of Fe_2_O_3_ while assuming that the percentage catalyst content reduces the biomass content in the fuel by the same percentage. An analysis of the carbon dioxide content in the flue gases from the combustion of wood and furniture board pellets is presented in Table 6.

After comparing the calculated and experimentally obtained results, it can be concluded that the calculated values are close to the experimental ones, which means that the catalyst content did not affect the combustion process.

Analysis of the effect of Fe_2_O_3_ content on gas emissions: CO, NO_x_, and H_2_ based on Figure 4 showed a clear decrease in the hydrogen content of the flue gas during the combustion of wood pellets only at 15% catalyst content, and in the case of slab pellets at 10% catalyst content. This is applied to both wood pellets and furniture board pellets. There was also a noticeable effect of the catalyst content of the fuel on NO_x_ emission values.

To verify whether the catalyst used had a significant impact on reducing the emissions of such gases as CO, NO_x_, and H_2_ in the original version of the emission results, the results of the content of these gases recorded during the experiments were divided by the results of CO_2_ contents, taking this quantity as a measure of the intensity of the combustion process. After a statistical analysis and an analysis of variance, the results obtained are presented in Figure 5. The trend lines for the emission values are given in the graphs, and the calculated values of the coefficients are shown in Table 7.

An analysis of the graphs in Figure 5 and the regression equations in Table 7 indicates that adding Fe_2_O_3_ iron oxide to the biomass fuel reduces emissions of the following compounds produced during fuel combustion: NO_x_, CO, and H_2_. However, the positive effect of the catalyst on reducing hydrogen emissions occurred only at a content of 15% Fe_2_O_3_ in the fuel. The impact of the catalyst on NO_x_ emissions is significantly more substantial with furniture boards used as fuel than wood. There is a negligible reduction in CO emissions in wood pellet and furniture board combustion.

### 3.3. Temperature Analysis

A statistical study of the results of the temperatures obtained over the burner and the temperature of the gases in the duct leading these out of the boiler is shown in Figure 6.

**Figure 6 materials-15-07740-f006:**
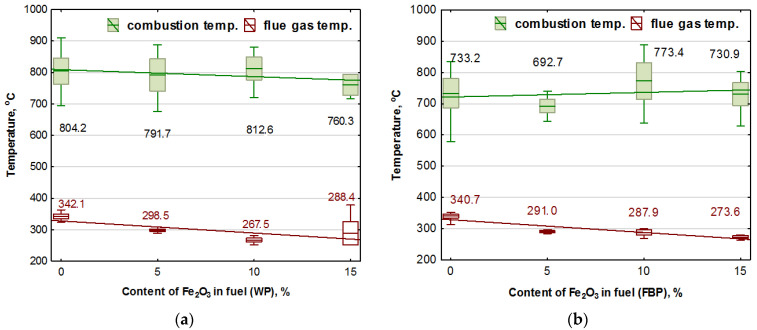
Influence of Fe_2_O_3_ content in the fuel on the combustion temperature and boiler flue gas temperature during the combustion of wood pellets (WP) and furniture board pellets (FBP): (**a**) Wood pellets (WP); and (**b**) Furniture board pellets (FBP). The graph shows one combustion temperature for each type of pellet, calculated as the average results obtained from the three thermocouples placed above the burner. The regression equations for the trend line and the calculated values for the value of the p coefficient determined as a result of an ANOVA analysis are shown in Table 8.

**Table 8 materials-15-07740-t008:** Regression equations for the trend line of the dependence of pellet combustion temperature and exhaust gas temperature on the Fe_2_O_3_ content of the combusted pellet.

Temperature	Wood Pellet		Furniture Board Pellet	
	Regression Equation	*p*-Value	Regression Equation	*p*-Value
Combustion temp., °C	y = 808.8 − 2.2x	0.0000	y = 721.5 + 1.5x	0.0000
Flue gas temp., °C	y = 327.9 − 3.8x	0.0000	y = 328.9 − 4.1x	0.0000

By analysing the trend lines, it was found that the combustion temperature of the pellets containing wood decreased with the participation of the catalyst in the fuel, and the combustion temperature of the granulate from furniture board waste increased. By analysing the mean values of the combustion temperatures of the fuel and their standard deviations, it was noted that the combustion temperature remained basically at the same level, regardless of the catalyst content.

The fuel combustion temperature was the resultant flame temperature. The flue gas temperature was the result of the heat generated from burning the fuel pellets and the heat that the flue gas gave up to the water jacket. As the amount of the fuel decreased with the increase in catalyst content, the amount of heat generated from burning the pellets was also reduced. In the case of the flue gas temperature, this was a downward trend line.

The average combustion temperature of each type of pellet in each case was below 900 °C, and its average value oscillated between 800 and 700 °C, so this was not the iron smelting temperature [36]. Oxygen reduction in the catalyst at this temperature was not very intensive [38], so its effect on the combustion process and reducing carbon monoxide and hydrogen were not very high, either. The low combustion temperature also decreased the group of nitrogen oxides of thermal origin.

After the study, the chemical and physical parameters of the combustion residues taken from the boiler ash pan were analyzed. The content of individual chemical compounds in the ashes from the combustion of wood pellets and furniture board pellets, depending on the proportion of the catalyst, is listed in Table 9.

Fe_2_O_3_ was the dominant chemical compound in the combustion residues of fuels containing iron oxide. A small amount of Fe_2_O_3_ was also found in the ashes from pellet combustion without adding a catalyst. This proportion of iron oxide is due to the natural composition of the raw material used in the base fuel.

The analysis of ash content and unburned fuel residues in ashes from the combustion of both types of pellets is presented in Table 10. In order to be able to assess the amount of unburned fuel in the residue material from the combustion of pellets without Fe_2_O_3_ and with this catalyst, the theoretical amount of organic matter that was burned to give an ash content identical to that in Table 10 was determined. For this purpose, the ash content of both materials (WP and WBP, respectively) from Table 3 was used. The Fe_2_O_3_ content of the pellet was also included in the calculation as the mineral component of the ash. The ratio of the unburned fuel (Total fuel of Ash from Table 10) to the total calculated organic matter contained in the burned corresponding type of pellet was then determined. The results of the calculations are also shown in Table 10 in the last column.

A comparative analysis of the ratio of unburned fuel in the ashes from pellet combustion with and without Fe_2_O_3_ content also indicates that the addition of a catalyst to the fuel pellet may have influenced changes in combustion quality but in a non-significant way. There was a slight increase in the unburned organic matter when burning wood pellets with Fe_2_O_3_ content compared to pellets without iron oxide (from 0.09% to 0.22%). There were also small changes in the ratio of dry organic matter in the residue from the combustion of pellets made from furniture board without iron oxide to the residue from the combustion of a similar pellet with an amount of this mineral (0.50 to 0.31%). In this case, there was a slight reduction in the amount of unburned fuel in the pellet combustion residues.

It is natural to have small amounts of unburned fuel in the residue from the combustion of solid fuels in boilers. During the combustion of fuels in heating boilers, there is also natural for this process to be disrupted, manifesting as temporary increases in the hydrogen and carbon monoxide content of the flue gases emitted (Figure 5b).

It was impossible to determine from the Fe_2_O_3_ content of the remaining ash from combustion whether oxygen reduction occurred in the catalyst used and how it affected the emission results, as reoxidation of free iron may have occurred as the ash cooled. The oxygen reduction effect of the iron oxide may have happened during hydrogen and carbon monoxide reduction in the flue gas.

## 4. Conclusions

It was not found that the addition of the Fe_2_O_3_ catalyst to pellets containing furniture board waste or wood caused any deterioration in the combustion quality of the fuel. This fact was confirmed based on an analysis of carbon dioxide emissions and the composition of the ash.

It was found that the addition of Fe_2_O_3_ to the biofuel had a positive effect on reducing hydrogen and nitrogen oxide emissions. A reduction in the content of nitrogen oxides, with the addition of Fe_2_O_3_ to the fuel, occurred in the combustion of pellets made from spruce wood and furniture board waste. The reduction of hydrogen and nitrogen oxide emissions in the flue gas with Fe_2_O_3_ was much more substantial when burning pellets containing furniture board waste than when burning pellets containing wood. This effect was mainly due to the high concentration of NO_x_ emissions in the flue gas resulting from the combustion of materials with high nitrogen content.

## Figures and Tables

**Figure 1 materials-15-07740-f001:**
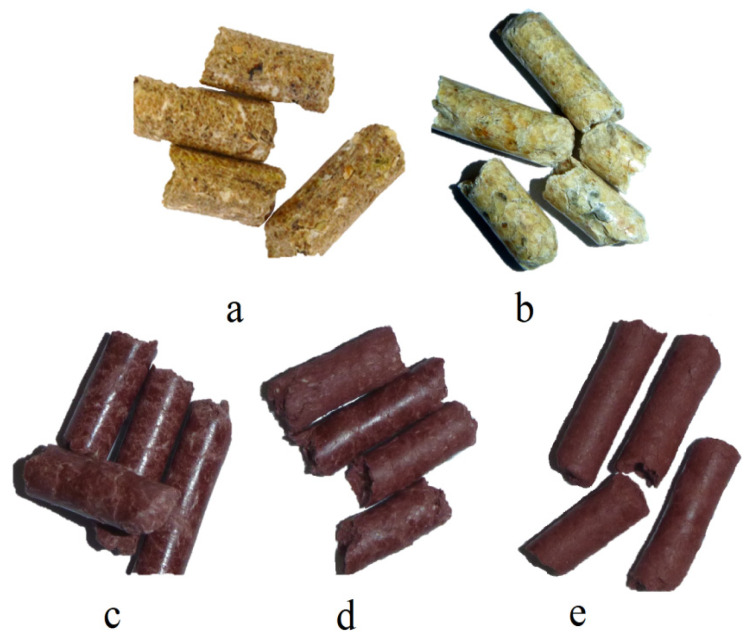
Examples of pellets: (**a**) From shredded furniture boards only; (**b**) From spruce wood only; (**c**) 5% Fe_2_O_3_ content; (**d**) 10% Fe_2_O_3_ content; and (**e**) 15% Fe_2_O_3_ content.

**Figure 2 materials-15-07740-f002:**
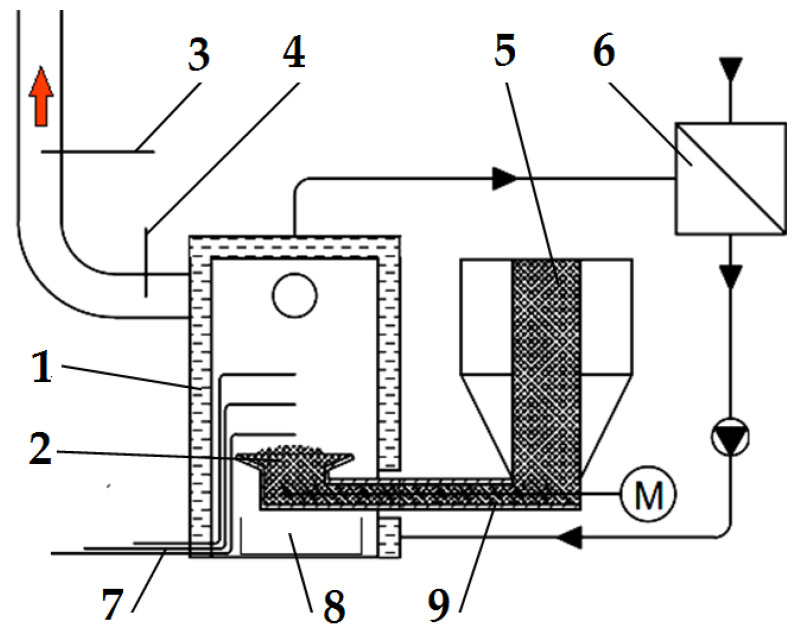
Schematic diagram of the emission measurement test bench: (1) boiler, (2) retort burner, (3) flue gas analyser probe, (4) thermocouple for measuring flue gas temperature, (5) fuel dosing line, (6) heat exchanger, (7) thermocouples for measuring flame temperature, (8) ash container, and (9) screw conveyor. Red arrow—flue gas, M—screw conveyor motor.

**Figure 3 materials-15-07740-f003:**
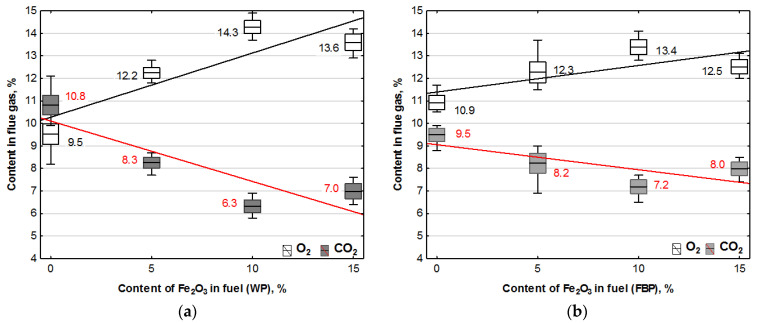
Impact of Fe_2_O_3_ content in fuel on oxygen and carbon dioxide contents in flue gases from the combustion of: (**a**) Wood pellets (WP); and (**b**) Furniture board pellets (FBP).

**Figure 4 materials-15-07740-f004:**
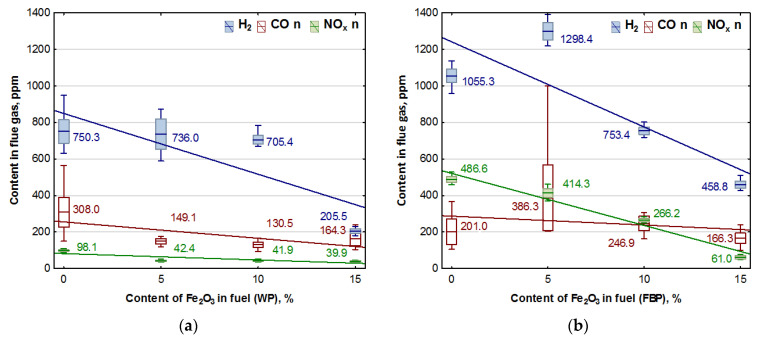
Impact of Fe_2_O_3_ content in fuel on hydrogen, carbon monoxide, and NO_x_ contents in flue gases from the combustion of: (**a**) Wood pellets (WP); and (**b**) Furniture board pellets (FBP).

**Figure 5 materials-15-07740-f005:**
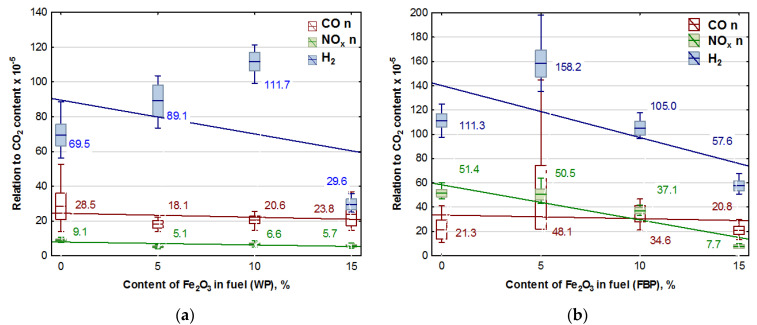
Graphs of CO, NO_x_, and H_2_ content in flue gases (coordinated to CO_2_ emission values) in dependence on the Fe_2_O_3_ content in pellets: (**a**) Wood pellets (WP); and (**b**) Furniture board pellets (FBP).

**Table 1 materials-15-07740-t001:** Specification of the Wöhler A 550 analyser [40].

Component	O_2_	CO_2_	CO	NO_x_
Measurement principle	Electrochemical sensor	Nondispersive infrared sensor-NDIR	Electrochemical sensor, H_2_ compensated	Measurement principle
Range	0–21 vol.%	0–40 vol.%	0–4000 vol. ppm	0–1000 vol. ppm (continuously up to 200)
Accuracy	±0.3 vol.%	0–6 vol.%: ±0.3 vol.%, 6–40 vol.%: ±5% of reading	±5 vol. ppm (<100 ppm), otherwise 5% of reading	±5 vol. ppm (<100 ppm), otherwise 5% of reading

**Table 2 materials-15-07740-t002:** Elemental composition of prepared base products.

Biofuel Base	Mass Content, % (Dry Mass)				
	Carbon	Hydrogen	Nitrogen	Sulfur	Oxygen
Wood pellets (WP)	50.50 ± 0.92	5.95 ± 0.20	0.11 ± 0.01	0.00 ± 0.00	43.14 ± 0.46
Furniture boards pellets (FBP)	48.96 ± 1.01	5.94 ± 0.52	4.25 ± 0.34	0.02 ± 0.02	39.43 ± 0.50

**Table 3 materials-15-07740-t003:** Fuel composition of prepared base products.

Biofuel Base	Mass Content, % (Dry Mass)			Combustion Heat, kJ·kg^−1^
	Ash	Volatile Fuel	Solid Fuel	
Wood pellets (WP)	0.3 ± 0.06	81.80 ± 0.95	17.90 ± 0.89	20,553 ± 200
Furniture boards pellets (FBP)	1.40 ± 0.50	77.80 ± 0.40	20.80 ± 0.20	19,960 ± 236

**Table 4 materials-15-07740-t004:** Pellet moisture content.

Biofuel with Addition Fe_2_O_3_	Moisture Content, %			
	0 Fe_2_O_3_	5 Fe_2_O_3_	10 Fe_2_O_3_	15 Fe_2_O_3_
Wood Pellets (WP)	8.5 ± 0.2	12.7 ± 0.1	11.6 ± 0.1	12.9 ± 0.2
Furniture Boards Pellets (FBP)	8.9 ± 0.1	13.1 ± 0.2	11.6 ± 0.3	11.5 ± 0.2

**Table 5 materials-15-07740-t005:** Equations describing trend lines and the values of *p*-coefficient.

Emission Content	Wood Pellet		Furniture Board Pellet	
	Regression Equations	*p*-Value	Regression Equations	*p*-Value
O_2_, %	y = 10.27 + 0.29x	0.0000	y= 11.39 + 0.12x	0.0000
CO_2_, %	y = 10.11 − 0.27x	0.0000	y = 9.06 − 0.11x	0.0000
H_2_, ppm	y = 849.03 − 33.30x	0.0000	y = 1241.64 − 46.69x	0.0000
CO n, ppm	y = 255.43 − 9.00x	0.0000	y = 286.65 − 4.87x	0.0000
NO_x_ n, ppm	y = 81.84 − 3.50x	0.0000	y = 520.77 − 28.50x	0.0000

Where: x–value of Fe_2_O_3_ content in fuel, %.

**Table 6 materials-15-07740-t006:** Analysis of carbon dioxide content in flue gases.

Based Material Content in Pellet, %	Wood Pellet				Furniture Board Pellet			
	100.0	95.0	90.0	85.0	100.0	95.0	90.0	85.0
CO_2_ content determined experimentally (Figure 3)	10.8	8.3	6.3	7.0	9.5	8.2	7.2	8.0
CO_2_ content calculated theoretically based on the proportion of fuel in the pellet	11.2	8.5	6.6	7.0	9.6	8.2	7.2	8.1
Emission differences between determined and calculated content	−0.4	−0.2	−0.3	0.0	−0.1	0.0	0.0	−0.1

**Table 7 materials-15-07740-t007:** Regression equations for the trend line of the dependence of CO, NO_x_, and H_2_ emissions from CO_2_ content and *p*-values.

Relation to CO_2_ Content × 10^−5^	Wood Pellet		Furniture Board Pellet	
	Regression Equation	*p*-Value	Regression Equation	*p*-Value
H_2_,	y = 89.5243 − 1.943x	0.0000	y = 140.1937 − 4.2885x	0.0000
CO n,	y = 24.4734 − 0.2303x	0.0000	y = 33.4648 − 0.3021x	0.0000
NO_x_ n,	y = 7.9341 − 0.1711x	0.0000	y = 58.3273 − 2.8884x	0.0000

**Table 9 materials-15-07740-t009:** Contents of individual chemical compounds in residues from the combustion of wood pellets and furniture board pellets, depending on the proportion of the catalyst in the pellets.

Pellet Composition	Mass Content, %											
	CaO	Fe_2_O_3_	P_2_O_5_	SiO_2_	K_2_O	MgO	SO_3_	Cl	Al_2_O_3_	MnO	TiO_2_	Ash
WP 0% Fe_2_O_3_	48.89	2.05	1.79	2.13	6.78	2.15	1.35	0.1	0.37	5.99	0.22	71.82
WP 5% Fe_2_O_3_	3.32	87.07	0.46	0.76	1.97	0.21	0.27	0.08	0.07	2.20	0.04	96.45
WP 10% Fe_2_O_3_	2.51	89.40	0.40	0.51	1.35	0.1	0.20	0.10	0.10	1.50	0.08	96.25
WP 15% Fe_2_O_3_	1.75	92.20	0.30	0.23	0.79	0.0	0.16	0.11	0.00	0.95	0.16	96.65
FBP 0% Fe_2_O_3_	52.20	2.84	1.42	3.2	6.25	0.35	0.81	0.05	0.70	4.18	2.42	74.42
FBP 5% Fe_2_O_3_	4.48	86.25	0.45	0.93	1.99	0.26	0.37	0.00	0.00	0.15	0.49	95.37
FBP 10% Fe_2_O_3_	3.44	88.68	0.34	0.83	1.10	0.0	0.24	0.06	0.00	0.40	0.43	95.52
FBP 15% Fe_2_O_3_	1.88	92.32	0.20	0.55	0.82	0.05	0.21	0.04	0.00	0.17	0.41	96.65

**Table 10 materials-15-07740-t010:** Composition of ashes from the combustion of wood pellets (WP) and furniture board pellets (FBP) depending on the catalyst proportion (converted into dry mass).

Pellet Composition	Mass Content, % (Dry Mass)			Solid Fuel	Ratio of the Unburned Organic Matter to the Total Organic Matter Contained in the Combusted Pellet, %
	Ash	Total Fuel of Ash	Volatile Fuel		
WP 0% Fe_2_O_3_	77.84	22.15	20.54	1.62	0.09
WP 5% Fe_2_O_3_	97.29	2.71	2.34	0.37	0.16
WP 10% Fe_2_O_3_	98.16	1.84	1.70	0.14	0.21
WP 15% Fe_2_O_3_	98.78	1.22	1.22	0.00	0.22
FBP 0% Fe_2_O_3_	73.98	26.02	24.49	1.53	0.50
FBP 5% Fe_2_O_3_	96.74	3.26	2.12	1.14	0.23
FBP 10% Fe_2_O_3_	97.71	2.29	1.29	1.00	0.30
FBP 15% Fe_2_O_3_	98.40	1.60	1.60	0.00	0.31

## Data Availability

The data presented in this study are available on request from the corresponding author.

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
