# Peer review of "Analysis of the Effect of Fe2O3 Addition in the Combustion of a Wood-Based Fuel"

_materials, 2022, doi:10.3390/ma15217740_

Round 1

Reviewer 1 Report

The article titled Analysis of the Effect of Fe2O3 Addition in the Combustion of a Wood-Based Fuel” studies the flue gas composition of combusted furniture board waste and wood pellets mixed with Fe2O3. The presence of Fe2O3 reduced the concentration of hydrogen and nitrogen oxide emissions in the flue gas is very informative. The introduction covered the background literature and experimental condition are clearly mentioned. The article is good and conclusive and can be accepted with minor modifications.

Comments:

In table 2 the Elemental composition of WP and FBP shows the mass% of ~56 and ~59%, how do the authors account for the remaining? If ash was added to this, then the values raise to ~57 and ~61%. Need a proper justification and explanation. Same with Table 10.

Author Response

The authors would like to thank the reviewer for his encouraging comments and for noticing and passing on certain inaccuracies in the tables provided. Below are our responses.

In table 2 the Elemental composition of WP and FBP shows the mass% of ~56 and ~59%, how do the authors account for the remaining? If ash was added to this, then the values raise to ~57 and ~61%. Need a proper justification and explanation. Same with Table 10.

Answers:

For Table 2, the proportion of oxygen in the elemental composition of the base materials used in the study was omitted. Table 2 has been completed with the missing element.

The data in Table 10 have been corrected. The only excuse for these errors were mistakes. Thank you again.

Reviewer 2 Report

The manuscript titled as “Analysis of the Effect of Fe2O3 Addition in the Combustion of a Wood-Based Fuel” seems a comparative study was carried out concerning emissions arising from a catalytic 15 combustion of pellets made from furniture board waste and wood mixed with Fe2O3. Though, manuscript is well drafted and contains lots of results for scientific community. But, the present condition of the manuscript warrants some modification for better readability of the researchers. The suggestions are as follows:

1.      Some more quantitative results must be shown in abstract.

2.      A lot of recent work in the same area has been published in 2022, kindly refer and discuss them in introduction section.

3.       The section “2.2.1. Fuel and residues after combustion assessment methods” contain some non-significant information, authors are suggested to crisp this section and keep only significant information.

4.      What about the density of each variety of pallet? Have authors calculated that? Is there any influence of that on the combustion?

5.      “2.2.1. Flue gas emission test bench”……….check the section numbering…………

6.      Result section is well written, kindly compare your results with some of published work

7.      Conclusion: Better to keep point wise (max 6)crisp information in max two lines.

Author Response

The authors thank the reviewer for the many valuable comments on the manuscript and provide the following responses.

  1. Some more quantitative results must be shown in the abstract.

The following sentences have been added to the abstract:

The average flame temperature in boiler was between 730 and 800°C. Line 18

A strong effect of the addition of Fe2O3 on the reduction of the average NOx content in the flue gas occurred with the combustion of furniture board fuel, from 51.4 ppm at zero Fe2O3 range to 7.7 ppm additive content of 15%. Lines 25‒27

As a minor meaningful sentence has been removed instead:

The additive of Fe2O3 impacted the reduction of CO2 emissions, but only due to a reduction in the combustible mass content of both wood pellets and furniture board waste pellets.

  1. A lot of recent work in the same area has been published in 2022, kindly refer and discuss them in introduction section.

The introduction section has been enlarged to include several new items, as recommended by the reviewer. These are items in references 19,24, and 35. If the reviewer feels that a relevant publication has been omitted, the authors ask for its details.

  1. The section “2.2.1. Fuel and residues after combustion assessment methods” contain some non-significant information, authors are suggested to crisp this section and keep only significant information.

The section “2.2.1. Fuel and residues after combustion assessment methods” has been reviewed for the relevance of the information contained in this section to the description in the manuscript of the experiments conducted and the significance of the information contained therein to the readers of our article. The information contained herein may be less relevant to experienced scientists conducting similar research. The readers of this article may also be undergraduate and postgraduate students to whom the information may be highly relevant. It was therefore considered that this section would not be changed.

  1. What about the density of each variety of pellet? Have authors calculated that? Is there any influence of that on the combustion?

The authors thank the reviewer for the interesting and insightful question. In this research paper, we only reported the density and bulk density of Fe2O3 added to its production, lines: 127‒130. We have only assessed the pellet made from the base material in terms of its moisture elemental composition and combustible part content. In this study, we did not consider the effect of pellet density on combustion, but we will consider this issue in future experiments.

  1. “2.2.1. Flue gas emission test bench”……….check the section numbering…………

The section number has been corrected

  1. Result section is well written, kindly compare your results with some of published work

It was only possible to compare the results in our manuscript with those of published papers where there was some similarity in the research carried out. General references to published articles are cited in section "3.3. Temperature analysis". Lines: 376 and 377

  1. Conclusion: Better to keep point wise (max 6)crisp information in max two lines.

The authors thank the reviewer for his comments on editing the “Conclusions” section, which we agree with and will take into account in our future work.

Reviewer 3 Report

The manuscript reported the influence of Fe2O3 on the combustion of a wood based fuel, and the comments are listed below.

(1) Why the dosage of Fe2O3 was used 0, 5% 10% and 15%?

(2) Please give more details that addition of Fe2O3 could decrease on the  NOx, CO, and H2 emissions.

(3) Fe2O3 could reduce the NOx and CO , how about the other oxides such FeO Fe3O4 or Fe ?

Author Response

Response to reviewer comments

The authors thank the reviewer for his valuable comments on our manuscript and provide the following responses.

  • Why the dosage of Fe2O3 was used 0, 5% 10% and 15%?

The choice of a wide range of Fe2O3 ratios in the prepared fuels was based on the assumption that reduced iron oxide oxygen would assist the combustion process and on the lack of knowledge available in the literature on the catalytic action of iron oxide in reducing gas emissions from biomass combustion.

This explanation is also included in the text of the manuscript – lines 136 ‒140

  • Please give more details that addition of Fe2O3 could decrease on the  NOx, CO, and H2

Although no iron oxide FeO or free iron Fe was found in the cooled ash, a selective reduction of Fe2O3 to these substances and the formation of H2O from free hydrogen and CO2 from CO during fuel combustion cannot be excluded. The finding of only Fe2O3 in the cooled ash could be due to the secondary oxidation of Fe and FeO.

 (3) Fe2O3 could reduce the NOx and CO , how about the other oxides such FeO Fe3O4 or Fe ?

The instruments and measuring methods used in the study did not reveal the existence of free iron and FeO, and it can only be assumed that their formation was possible despite the low combustion temperature. At the same time, too low a temperature during biomass combustion prevented the formation of Fe3O4.

A general answer to the reviewer's questions (2) and (3) is provided in the text of section “3.3 Temperature analysis”, lines 374 ‒379

Round 2

Reviewer 3 Report

The authors revised the manuscript according to the comments well, and the economic analysis should be discussed if this technology of Fe2O3 Addition in the Combustion of a Wood-Based Fuel for the industrial application. 

Author Response

The authors would like to thank the reviewer very much for highlighting the economic aspect of using the Fe2O3 additive to biofuel to reduce the emissions produced during its combustion.

Our manuscript is concerned with the technical and energy analysis of the materials used in the study. This task would far exceed the substantive scope of this manuscript because performing a sound economic analysis of the technology we used and discussing its cost-effectiveness would require additional economic studies.

The authors plan to conduct an economic evaluation of the use of catalytic fuel additives in the future but in the context of using a greater variety of such additives and based on more research.